# Assessment of the Efficiency of Technological Processes to Modify Whey Protein Antigenicity

**DOI:** 10.3390/foods12183361

**Published:** 2023-09-07

**Authors:** Vanina Andrea Ambrosi, Silvina Mabel Guidi, Debora Marina Primrose, Claudia Beatriz Gonzalez, Gustavo Alberto Polenta

**Affiliations:** 1Instituto Nacional de Tecnología Agropecuaria (INTA), Instituto Tecnología de Alimentos, CC 25, Castelar CP 1712, Argentina; 2Instituto de Ciencia y Tecnología de Sistemas Alimentarios Sustentables, UEDD, INTA, CC 25, Castelar CP 1712, Argentina; 3Facultad de Farmacia y Bioquímica, Universidad de Buenos Aires (UBA), Junín 954, Buenos Aires C1113AAD, Argentina; 4Escuela Superior de Ingeniería, Informática y Ciencias Agroalimentarias, Universidad de Morón, Cabildo 134, Morón B1708WAB, Argentina; 5National Council of Science and Technology (CONICET), Godoy Cruz 2290, Buenos Aires C1425FQB, Argentina; 6Instituto de Biotecnología, Universidad Nacional de Hurlingham (UNAHUR), Av Vergara 2222, Hurlingham CP 1688, Argentina

**Keywords:** enzymatic hydrolysis, high hydrostatic pressure (HHP), allergens, ELISA, β-lactoglobulin (BLG)

## Abstract

Whey is a by-product that represents a cheap source of protein with a high nutritional value, often used to improve food quality. When used as a raw material to produce hypoallergenic infant formulas (HIF), a processing step able to decrease the allergenic potential is required to guarantee their safe use for this purpose. In the present paper, thermal treatments, high hydrostatic pressure (HHP), and enzymatic hydrolysis (EH) were assessed to decrease the antigenicity of whey protein solutions (WPC). For monitoring purposes, a competitive ELISA method, able to detect the major and most allergenic whey protein β-lactoglobulin (BLG), was developed as a first step to evaluate the efficiency of the processes. Results showed that EH together with HHP was the most effective combination to reduce WPC antigenicity. The evaluation method proved useful to monitor the processes and to be employed in the quality control of the final product, to guarantee the efficiency, and in protein antigenicity reduction.

## 1. Introduction

Food allergies are a major food safety concern in industrialized countries, affecting up to 4% of adults and 5% of children [1]. Dairy products are one of the “Big 8” foods that are responsible for 90% of severe allergic reactions [2]. Cow milk allergy is the most common food allergy in infants. Children with this condition need to consume replacement products, known as hypoallergenic infant formulas (HIF), which are made of proteins that have been extensively hydrolyzed to reduce their immunologic reactivity.

According to clinical and regulatory requests, hypoallergenic formulas should not cause symptoms in 90% of sensitive infants who are fed these products [3]. Despite this stringent requirement, several allergy episodes have been reported, involving commercial HIF, which allegedly occurred when products with relatively large remaining peptides were consumed by the most sensitive children [4]. The main causes of these episodes were incomplete hydrolysis due to the inaccessibility of proteases to some epitopes or the exposure of originally buried antigenic epitopes caused by different processes [5].

To control the compliance with this requirement, immunological methods are the most useful tools. ELISA methods can be specifically developed to ensure that these quality specifications are met. However, since commercial kits are developed to detect and quantify milk traces, using these kits for monitoring purposes would require high serial dilutions of the samples to achieve the optimal measurement range.

In the case of HIF obtained from allergenic sources such as milk, the first step is to identify an efficient processing technology to reduce the allergenicity of the raw material and rigorously evaluate its practical feasibility. Heat treatments may not be adequate as a sole processing step, as they can randomly affect the antigenicity by modifying the conformation of proteins. Indeed, several studies have shown that heat denaturation of β-lactoglobulin (BLG) can also expose buried epitopes, rendering the so-called neoallergens [6].

Other technologies such as enzymatic hydrolysis (EH) would be preferred because of their simplicity and low cost, representing an economic alternative to modifying protein antigenicity [7,8]. In fact, whey hydrolysates have been used for a long time in infant nutrition as substitutes for human breast milk [9]. The selection of hydrolysis methods mainly depends on the protein source. For example, keratin sources such as feathers, horns, and beaks usually require acidic or alkaline hydrolysis treatments, or the use of bacterial keratinases. On the other hand, the proteolysis of animal products such as whey and meat, or plant ingredients such as soy and legume proteins, often requires enzymatic or microbial hydrolysis [10]. In this regard, different peptidases also allow the production of oligopeptides with different biological properties. The identification of new enzymes, especially of microbial origin, able to act with different specificities over different protein sources represents an active field of research [11].

In recent years, the use of different emerging processes to reduce allergenicity has also been proposed. Among them, high hydrostatic pressure (HHP) represents a potentially useful technology, as it can affect the tertiary and quaternary structure of proteins. However, since the primary structure of proteins would remain intact, sequential epitopes are expected to be rather unaffected by these treatments [12]. Previous studies have shown that pressures above 300 MPa can cause irreversible changes, leading to the denaturation of the protein, although the protein core is still able to resist pressures of up to 400 MPa [13,14]. A promising approach to enhance the effectiveness and accelerate the process is to combine both technologies by conducting the hydrolysis under the assistance of HHP [14,15,16]. The combined application of the two processes would induce the exposure of cleavage sites inaccessible to proteases under regular conditions [17].

As mentioned above, a suitable ELISA method should be available to effectively measure the remaining antigenicity of specific milk proteins for monitoring the process. This is a relevant component needed to compose a technological package for HIF production. BLG can be considered as an adequate target for developing such an analytical tool, as it is the major whey allergen (10% of milk proteins, 50% of the whey proteins). This protein has a molecular weight of 18.3 kDa with 162 amino acid residues, and can be found in either monomeric or dimeric conformation [18]. Under physiological conditions (neutral pH and BLG concentration >50 µM), BLG is predominantly dimeric, with about 6% of the monomer surface area buried in the protein–protein interface [19].

The aim of this present study was to evaluate and optimize different technologies, alone or in combination, to produce HIF by using an in-house-developed ELISA method as a quality control tool to monitor the antigenicity of BLG.

## 2. Materials and Methods

### 2.1. Raw Material

Whey protein concentrate (WPC-80) Lacprodan^®^ 80 from Arla Foods Ingredients, Argentina was used as substrate for the evaluation of the treatments. Protein content was quantified by the Kjeldahl method [20]. WPC-80 contains 78.3 ± 0.5% of total protein. A 1% (*w*/*v*) WPC-80 solution was prepared with distilled water, mixed for 1 h and then stored at 4 °C until use.

### 2.2. Enzyme

Bromelain from pineapple stem (B 4882 3 to 7 U/g) was purchased from Sigma Aldrich, Germany, prepared at a concentration of 10 caseinolitic units/mL in sodium phosphate buffer (877 mM NaH_2_PO_4_, 123 mM Na_2_HPO_4_, pH 6), and kept refrigerated at 4 °C until use.

### 2.3. Degree of Hydrolysis

Degree of hydrolysis (DH) is defined as the proportion of cleaved peptide bonds in a protein hydrolysate [21]. The increase in %DH provoked by the hydrolysis was assessed by means of derivatization with OPA (o-phthaladehyde) in the presence of 2-mercaptoethanol (thiol agent), which reacts quantitatively with primary amines forming adducts that strongly absorb at 340 nm [22]. The OPA reagent was prepared by combining 10% of OPA (50 mM in methanol), 10% of 2-mercaptoethanol, 5% SDS (20% *w*/*v*), and 75% of borate buffer (0.1 M Na_2_B_4_O_7_·10H_2_O, pH 9.5). OPA reagent was protected from light with aluminum foil, and allowed to stir for at least 1 h before use. The OPA assay was carried out with some modifications as described by Spellman et al. [23]. Twenty milliliters of sample (or standard) were added to 2.4 mL of OPA reagent. The absorbance was measured at 340 nm with a VIS Spectrophotometer (Metrolab 330, Metrolab OEM, Quilmes, Argentina) after 10 min. DH of 0% was assigned to untreated 1% WPC-80 (control).

Percentage of DH values was calculated by using Equation (1):DH(%) = (∆Abs 1.934d)/c(1)
where ∆Abs is Abs340 of treated samples minus Abs340 of untreated whey solution, d is dilution factor and c is protein concentration (g/L) [23].

### 2.4. Indirect Competitive ELISA

#### 2.4.1. Plates Preparation 

The Elisa plates (96 well, CostarStripwell ™ Plates, Flat bottom, High Binding, Clear, Poliestyrene microtitration wells, Corning incorporated, New York, NY, USA) were coated with 200 μL per well of 2 μg/mL BLG solution (L3908, Sigma Aldrich, Schnelldorf, Germany) in sodium carbonate–bicarbonate buffer (15 mM Na2CO3, 35 mM NaHCO_3_, pH 9.6). After incubation overnight at 4 °C, the wells were washed four times with washing solution (0.9%NaCl, 0.0125% Tween^®^20). Residual free binding sites were blocked with 250 μL of blocking buffer (1% (*p*/*v*) gelatine from cold water fish skin (G7041, Sigma Aldrich, Burlington, MA, USA), 0.05% (*v*/*v*) Tween^®^20 in TBS buffer (50 mM Tris, 150 mM NaCl, pH 7.4)), and incubated for 1 h at 37 °C. After washing four times with washing solution, plates were air dried and stored at −20 °C until use.

#### 2.4.2. Polyclonal Antibody 

A 10 mg/mL solution of BLG (L3908, Sigma Aldrich, Germany) was separated by electrophoresis on a bis-Tris SDS-PAGE system to obtain the purified protein in its denatured form. Protein bands were cut out and electroeluted to be used as immunogen. Polyclonal antibodies were raised in three New Zealand White rabbits at the Central Animal Laboratory of FCEyN-UBA, Argentina. Before starting the inoculations, control serum samples (pre-immune) were collected. For the immunization schedule, 1 mL of 1:1 complete Freund’s adjuvant containing 500 µg of protein (BLG) was injected subcutaneously at multiple sites in the back of each animal. Subsequent injections after 25 days were performed using Freund’s incomplete adjuvant containing 350 µg of the same immunogen. An exploratory bleeding at the back of the ear was performed after 10 days of the second dose of immunization to obtain whole serum, and in vitro immunoassay was performed to assess immune response. A third dose was then injected. For the final bleeding, animals were anesthetized, and cardiac puncture was performed. Immune sera were separated by centrifugation, pooled to conform a common polyclonal serum raised against denatured BLG, diluted in a 1:1 ratio in glycerol, and then stored at −20 °C until further use.

#### 2.4.3. Calibration Curve

A calibration curve was built by serial dilutions of WPC-80 in TBS. The range of concentrations was between 4 and 200,000 ng protein/mL

#### 2.4.4. Immunochemical Assay

For the competitive step of the ELISA assay, 100 μL of each standard/sample dilution plus 100 μL of anti-BLG antiserum obtained from rabbit (at a 1/1500 dilution in antibody buffer (3% (*p*/*v*) PEG, 0.1% (*v*/*v*) Tween^®^20 in TBS)) were added to the wells. Three replicates of each standard/sample were assayed by plate, in 2 different plates (6 replicates in total). Plates were incubated for 1 h at 25 °C on a rocking platform (Titramax 100 T, Heidolph, Germany) covered to avoid evaporation. Afterward, plates were emptied by inversion and washed four times with a washing solution. 

For the next step of the assay, 200 μL of AP-conjugated goat anti-rabbit IgG (#170-6518, BioRad, Hercules, CA, USA) diluted in 1/3000 in antibody buffer was loaded into each well, and incubated for 30 min at 25 °C. After four washings, a solution of 1 mg/mL of p-Nitrophenylphosphate substrate (P4744, Fulka, Sigma Aldrich, USA) in DEA buffer (0.1 M diethanolamine, 0.5 mM MgCl_2_, pH 9.8) was added in a final volume of 200 μL per well and incubated for 30 min at 25 °C, covered to avoid light and evaporation. The absorbance was determined at 405 nm using a microplate photometer reader (Packard Spectra Count™ BS10001, Los Angeles, CA, USA). Samples from different treatments taken to assess antigenicity were diluted (1:100 in TBS) to attain the range of the assay calibration curve.

#### 2.4.5. Antigenicity Measure

A four-parameters logistic model (4PL) was used to fit absorbance values to protein concentration of the standard solutions, using Equation (2):y = b + [(a − b)/(1 + (x/c)^d^)] (2)

This equation is used to express the data of immunoassays, where a is the maximum absorbance of the curve, b is the minimum absorbance, c is the concentration that provides 50% reduction in absorbance at half-maximal saturation (IC50), d is the slope of the curve, x is the concentration of either the standard or the sample, and y is the absorbance at 405 nm [24]. Analysis was performed with a software GraphPadPrism^®^ version 5.01 for Windows.

#### 2.4.6. Validation

##### Sensitivity

Linearity limits of the assay were determined by 3 assays, run on 5 different days. For the determination of the limit of detection (LOD) and the limit of quantification (LOQ), 24 measurements were taken in the absence of whey (at a concentration of 0 ng of whey protein/mL), which was considered as the maximum binding (B0). The LOD was defined as the lowest antigen concentration that can be detected from the background (B0), and is calculated with Equation (3) [25]. The LOQ is the lowest concentration at which the analyte can be reliably detected, and is calculated by subtracting ten times the standard deviation from the mean value of B0 and calculating the corresponding concentration from the equation of the respective calibration curve (Equation (4)) [26].
LOD = B0 − 3 SD_B0_(3)
LOQ = B0 − 10 SD_B0_(4)

SD_B0_ is the standard deviation of B0.

##### Precision

The intra-assay variability (repeatability or precision within a plate or run) was calculated by testing samples at low, medium, and high concentrations (95, 925 and 90,000 ng whey protein/mL) in 24 replicates each, on a single plate. The inter-assay repeatability or precision between plates and runs was calculated by running various dilutions in three different plates each run on 9 different days. The percent of coefficient of variation (%CV) for the intra-assay and inter-assay variability was determined by dividing the standard deviation of the replicates by the mean, then multiplied by 100.

##### Accuracy

For the recovery assays performed to evaluate the accuracy of the method, five samples with known concentrations of the calibrator were spiked in duplicate at the recovery values considered most relevant: an uncontaminated sample (0 ng/mL), a sample at a value close to the LOD (7 ng/mL), a sample at a value close to the LOQ (95 ng/mL), a sample at the value considered as acceptance, and a sample at a value close to 50% of the curve (925 ng/mL).

### 2.5. Whey Protein Treatments

Treatments applied to modify protein antigenicity include heat treatment, enzymatic hydrolysis, and HHP. The flow diagram of treatments applied to the WPC-80 solution is shown in Table 1.

#### 2.5.1. Control

The sample used as control was described in Section 2.1. In Table 1, it was assigned the code 1: C.

#### 2.5.2. Heat Treatment

Thermal treatment was applied when needed in order to inactivate the hydrolytic enzyme. Briefly, the tubes with the reaction mixture were placed in a water bath (Model Dubnoff, Vicking, Buenos Aires, Argentina) at 100 °C for 10 min and immediately cooled in an ice bath. Sample 2 corresponds to the control of the thermal inactivation treatment and was named 2: TIE.

#### 2.5.3. High Hydrostatic Pressure

HHP treatments were performed in a High pressure processor (Iso-Lab FPG9400:922, Stansted Fluid Power Ltd., Harlow, UK, 900 MPa Maximum pressure units, an inner volume of 2000 mL and working temperature range from −20 °C to 120 °C). WPC samples were packed in 110 mL one-way bottles, screw-capped, ensuring that there was no headspace left, sealed with Parafilm^®^ (PM-996, Pechiney, USA), and then double vacuum-sealed in polyethylene bags (Cryovac CN-640). Packed samples were first conditioned at 36 °C and then placed in the thermostatic pressure chamber (5 min). The pressure-transmitting medium was H_2_O:propylene glycol (70:30 *v*/*v*). Hydrostatic pressure was applied at 300 MPa and the built-up rate of pressurization and decompression was fixed at 300 MPa/min. A holding time of 5 min at 45 °C controlled temperature was fixed for HHP treatment. The temperature was controlled by K-type thermocouples. After decompression, samples were withdrawn from the pressure chamber and cooled in ice water. Sample 3 was used for comparative purposes to evaluate the effect of the sole application of HHP and was assigned the code 3: HHP.

#### 2.5.4. Enzymatic Hydrolysis

Hydrolysis with bromelain was carried out with an enzyme/substrate (E:S) ratio of 1:10 (*v*/*v*) at 45 °C constant temperature and 0.1 MPa (atmospheric pressure) for two periods of time, 15 or 30 min, in a thermostatic shaking water bath (Model Dubnoff, Vicking, Argentina). The flasks were then kept at RT for 10 min to simulate the time required for sample handling during HHP treatment. The hydrolysis reaction was stopped by applying TIE. Samples 4 and 6 correspond to hydrolysis (H) and extended hydrolysis (Hext), respectively.

#### 2.5.5. Combined Treatments 

##### Enzymatic Hydrolysis + High Hydrostatic Pressure

The hydrolysis treatment was applied as described in Section 2.5.4, except for the last period of 10 min at RT, which simulated the time required for the application of HHP treatment. To stop the hydrolysis, the sample was subjected to TIE, and thereafter, HHP treatment was applied (Section 2.5.3). This sample was named 5: H-In-HHP.

##### Enzymatic Hydrolysis Followed by High Hydrostatic Pressure

For this treatment, hydrolysis was first applied as described in Section 2.5.4 (without the 10 min period at RT) and immediately treated by HHP (Section 2.5.3). After sample removal from the pressurization chamber, the enzyme was inactivated by TIE. This treatment was designated as 7: H-HHP-In.

##### Enzymatic Hydrolysis Assisted by High Hydrostatic Pressure

For HHP-assisted hydrolysis, WPC samples containing the bromelain enzyme (E/S ratio 1/10 *w*/*w*) were packaged in plastic bottles, and immediately subjected to the HHP treatment (Section 2.5.3). After removal from the pressurization chamber, the enzyme was inactivated by TIE. This treatment was assigned the code 8: HHPaH.

### 2.6. Statistical Analysis

Experiments were performed in triplicate (3 independent runs). Standard deviations and mean standard error were calculated and reported on a case-by-case basis. The analysis of the variance tests was carried out using the InfoStat software version 2011. The effect of the treatments was statistically analyzed using the analysis of variance (ANOVA), followed by Tukey’s multiple comparison tests with a significance level *p* ≤ 0.05.

## 3. Results

### 3.1. Evaluation of the Performance of Indirect Competitive ELISA 

The ELISA was specifically designed and developed to estimate the remaining BLG antigenicity. This is one of the most important objectives of the study, as it can be used to evaluate the performance of the different treatments and the quality of the hydrolysates. Figure 1 shows the calibration curve, which was obtained by using a WPC-80 solution at different dilutions as a calibrator, and Table 2 presents the estimated parameters for the Four-parameters logistic model (4PL) regression model.

The method proved adequate for this purpose, with a suitable accuracy within the range of 4 and 200,000 ng whey protein/mL (R^2^ > 0.99).

### 3.2. ELISA Validation

#### 3.2.1. Sensitivity

The sensitivity of measurement methods is usually expressed by the LOD and LOQ values. The LOD is the lowest amount or concentration of the analyte in the test sample that can be reliably distinguished from zero, while LOQ is the smallest amount of analyte that can be accurately quantified [27]. Twenty-four measurements were taken in the absence of whey (B0). LOD and LOQ were calculated by using Equations (3) and (4). The LOD was found to be 6.6 ± 0.02 (ng whey protein/mL), and the LOD, 94.3 ± 0.05 (ng whey protein/mL).

#### 3.2.2. Precision

Assay precision was determined by its intra-and inter-assay repeatability. For the intra-assay repeatability, low, medium, and high concentration levels (90,925 and 90,000 ng whey protein/mL) were determined. A total of 24 replicates of each concentration was measured on a single plate. Thus, the intra-assay reproducibility with variation coefficients between 8.0 and 11.9% was determined. The inter-assay repeatability (precision between plates and runs) was calculated by running triplicates of calibration curves in three different plates each run on nine different days. The inter-assay variation raged between 7.53 and 12.5%.

#### 3.2.3. Accuracy

Five samples with known concentrations of the calibrator were spiked in duplicate at the recovery values considered most relevant: an uncontaminated sample (0 ng/mL), a sample at a value close to the LOD (7 ng/mL), a sample at a value close to the LOQ (95 ng/mL), a sample at the value considered as acceptance, and a sample at a value close to 50% of the curve (925 ng/mL). The recovery in all cases was between 70 and 135%, expressed as a percentage of the analyte. These results were consistent with similar studies [28].

Although different food commodities should be included in cross-reactivity testing for ELISA method validation, as stated by AOAC [29], this validation was not carried out in the present study because the use of the ELISA was not intended for the detection of milk traces in different food matrices.

Validation parameters for the ELISA method developed in this study are presented in Table 3.

One important decision to make in the development of the ELISA for this particular intended use is the election of the format. In this regard, competitive ELISA can be considered as the best alternative, since according to previous studies, it allows the increase in the detection range in comparison to the sandwich format [28]. The optimum working range estimated in the validation study is a significant achievement, as the content of the allergenic proteins in a hypoallergenic hydrolysate is typically within the range of 20 and 100 immunologically active μg antigen/g protein. Assuming that these ingredients are tested at a concentration of 10 mg protein/mL, the ELISA method should be sensitive in the range of <200 ng/mL [30]. Therefore, the results of this study demonstrate that the ELISA method is suitable for assessing and monitoring the remaining antigenicity during production, as well as for the quality control of the final product in the form of an infant formula.

Competitive ELISA is generally acknowledged as the best method to detect modifications of the immunoreactivity in proteins of interest such as BLG. This is because neither the DH nor the molecular mass distribution of hydrolysates properly reflect the degree of antigenicity reduction [31]. Factors such as the characteristics of antibody population, the ELISA format, and processing can influence the detectability of proteins such as BLG in processed food [32]. The most critical factor for the development of ELISA methods is the affinity of the antibody population used to recognize the target protein [33]. Some authors have highlighted that, if antibodies against one single protein are used, the target protein should be stable enough during food processing. An interesting strategy to improve the detection of allergens in processed foods is the use of antibodies raised against processed proteins (for instance, heated samples), which would permit a more efficient detection of allergens even in processed foods. In the present study, we used modified proteins to raise the antibodies, taking into account that the treatments to be applied are expected to induce protein denaturation. 

### 3.3. Effect of the Application of Combined High Hydrostatic Pressure Treatments and Enzymatic Hydrolysis in the Degree of Hydrolysis and in the Antigenicity of BLG in Whey Solution

Table 4 presents the effect of the combined application of enzymatic hydrolysis (bromelain) and HHP on a WPC solution. Results are expressed as %DH and BLG residual antigenicity (%BLG).

Control sample (1:C) corresponds to untreated WPC and was used as a basal reference. Therefore, %DH was arbitrarily considered as 0%, while the (residual) BLG antigenicity of this sample (%BLG) was 100%. The treatment 2:T-In, whereby the enzyme was inactivated by heat, induced a slight decrease in the %DH (*p* > 0.05), but this was not statistically significant. In turn, the %BLG significantly increased (*p* ≤ 0.05). Similar results were reported by Karamanova et al. [34] in milk subjected to heat treatments. Kleber et al. [35] suggest that in addition to the sequential or linear epitopes exposed in non-treated samples, buried epitopes can be exposed upon heating or other physical treatments. In agreement with this, treatment 3:HHP produced an effect similar to that observed in 2:T-In, also showing an increased antigenicity, with similar %BLG (*p* ≤ 0.05). 

The exposure of samples to high levels of pressure changed the antigenicity, probably through the denaturation and/or induction of molecular conformational changes in the proteins, causing the exposure of buried sites, which were previously inaccessible to antibodies [36,37]. Such changes were related to modifications in the tertiary structure and were detected after pressure treatments in the range 0–200 MPa. Between 200 and 400 MPa, further structural changes can be induced, although a faster refolding of proteins could take place at pressures closer to 200 MPa compared to 400 MPa [38].

The hydrolysis with bromelain (4:H-In) significantly modified both parameters, showing an increase of 4% in DH, and a reduction of around 13% in BLG (*p* ≤ 0.05), which would in turn reflect the reduction in the number of detectable epitopes induced by proteolysis.

It is known that the enzymatic hydrolysis of proteins, especially when digestive enzymes such as trypsin and/or chymotrypsin are used, is the most effective treatment to decrease allergenicity. This is because these enzymes can clive sequential epitopes [39], which are responsible for the allergic response. Therefore, enzymatic hydrolysis is the basis for the current industrial production of HIF [40]. 

The strategy of complementing the hydrolysis with heat treatment and HHP (5:H-In-HHP) showed no improvement in decreasing the antigenicity, as evidenced by the monitoring parameters (%DH and %BLG). The reduction attained in the antigenicity (17.6%) was similar to that of the treatment 4:H-In (13.4%; *p* > 0.05). This suggests that HHP has no effect on protein antigenicity once the protein has been hydrolyzed by the enzyme. These results are consistent with the effects observed in treatments 2:T-In and 3:HHP, and may be due to the fact that physical treatments such as HHP do not affect peptide bonds; and therefore, most of the epitopes remain unaffected [13]. Therefore, the reduction in the antigenicity would only occur during the first step of the treatment (cleavage of the epitopes normally exposed) with no further improvement by the heat and HHP applied thereafter [41,42].

In turn, treatment 6 (Hext-In extended proteolysis) showed a better performance than treatment 5 (H-In-HHP: hydrolysis followed by HHP) in terms of reduction in %BLG detectability. This indicates that HHP can improve the hydrolysis process, probably by enhancing the proteolytic process. The combination of both treatments (7:H+HHP-In) significantly reduced BLG antigenicity compared to treatment 6, which is likely due to the reduction in the number of available epitopes.

The combined treatment of hydrolysis followed by HHP and enzyme inactivation (7:H+HHP-In) significantly increased (*p* ≤ 0.05) the %DH compared to the extended hydrolysis (6:Hext-In). This is a strong evidence that the application of high pressures not only failed to stop proteolysis, but also improved the efficiency of the proteolytic process. 

Finally, the %DH produced by HHP-assisted hydrolysis (8:HHPaH-In) was not statistically different from treatment (7:H+HHP-In), even with a shorter treatment time. This can be considered an improvement in the process performance. This result demonstrates the synergistic effect of high pressures on the proteolytic process. High pressures likely induce the unfolding of proteins, expose hidden epitopes, and improve the effectiveness of the hydrolytic enzymes. As suggested in previous studies, hydrolysis carried out under a high pressure environment proves beneficial for the process performance [43,44,45,46].

The comparison of the residual antigenicity of the hydrolysates obtained by treatments 8:H+HHP-In and 7:HHPaH-In (57% and 80% reduction, respectively) with treatments 1 (C) and 6:Hext-In (27% and 50% reduction) clearly shows that high pressure assistance enhances the reduction in residual antigenicity. This effect is consistent with the increase in BLG digestibility observed after applying different pressure levels, where the exposure of proteins to high pressure promotes their unfolding; therefore, favoring the accessibility of digestive enzymes, as previously reported by Zeece et al. [47].

No significant difference in %DH was found between treatments 7 and 8 (*p* > 0.05), even though HHP can potentially induce relevant modifications in the components of the system, such as structural changes in proteins and inactivation of enzymes. However, it is important to note that treatment 8 (HHP-assisted hydrolysis) requires less time to achieve similar levels of %DH than treatment 7 (hydrolysis followed by HHP). Treatment 8 takes only 5 min of isothermal and isobaric treatment plus 10 min of sample manipulation, while treatment 7 takes 15 min for hydrolysis and an additional 15 min for high pressure exposure, for a total of 30 min. Therefore, from the industrial point of view (only considering the mentioned parameter as the evaluation criterion), HHP-assisted hydrolysis offers a higher level of economic feasibility.

A comparison of HHP-assisted hydrolysis (8) and hydrolytic process alone (6) shows that the high pressure can induce a greater %DH and render a product with lower antigenicity. This is probably due to the activation of enzymes and/or increased accessibility to previously buried sites, as a result of proteins unfolding. The lower antigenicity could be attributable to a more extended degradation of epitopic areas responsible for the immunological reactions [31], which was properly detected by antibody-based methodologies such as ELISA.

As an additional advantage, previous studies have shown that the application of an HHP treatment can reduce the formation of biogenic amines in dairy products. This reduction was linked to a significant decrease in microbiological counts, especially in the enterococci and enterobacteria groups [48]. Interestingly, in that study, proteolysis, which is mostly provoked by microbial enzymes (different to the case of commercial enzymes, as employed in our study) was prevented. This highlights the importance of testing the interaction of the treatment and enzyme for each intended use. 

Indeed, from the point of view of efficiency and economic feasibility of the proteolysis process, previous studies conducted by Garcia-Mora et al. [49] reported on the promising application of HHP for the cost-effective production of bioactive peptides of commercial interest by enzymatic proteolysis. These authors found this effect to be consistent for several commercial enzymes (Protamex, Savinase, and Corolase 7089), with an optimum pressure range between 100 and 300 kDa, which is coincident with our findings for pineapple bromelain.

The results of this study show that the developed BLG ELISA had a more effective performance than DH for detecting modifications of protein induced by different treatments. The selected method proved suitable for its intended purpose of providing a monitoring parameter to control the production of hydrolysates. It is also practical to use because it requires a low dilution of the sample. The antibodies obtained for analytical purposes allowed the development of a robust ELISA method that proved effective in assessing the antigenicity of BLG in the ranges required by the hypoallergenic specifications. Therefore, the methodology can be used to control all the stages of the manufacturing process of these types of products, from monitoring production to the quality control of the final product.

In practical terms, the fact that the residual antigenicity can be evaluated directly or with minimal dilution minimizes the error associated with the high levels of dilution that would be necessary with commercial kits. It is important to consider that the aim of the study was to assess the ability of ELISA methods to detect modifications of BLG in whey protein when different treatments are applied. However, it is important to bear in mind that even 20% residual antigenicity may include epitopes relevant for allergy patients. Therefore, further studies would be necessary to biologically validate this aspect. In this regard, we are currently conducting more sophisticated studies in our lab and in collaboration with other groups, including the use of animal models.

## 4. Conclusions

Among the different treatments evaluated, hydrolysis under high pressure was the most effective to reduce BLG antigenicity. This is a promising finding. Regarding another critical aspect also addressed in the present study, the ELISA method designed to detect the antigenicity of BLG submitted to different treatments with minor dilution proved suitable to measure and detect even slight antigenicity modifications quantitatively and accurately. It is a robust and convenient analytical tool that can be used to develop and control novel hypoallergenic food and hydrolysis processes.

## Figures and Tables

**Figure 1 foods-12-03361-f001:**
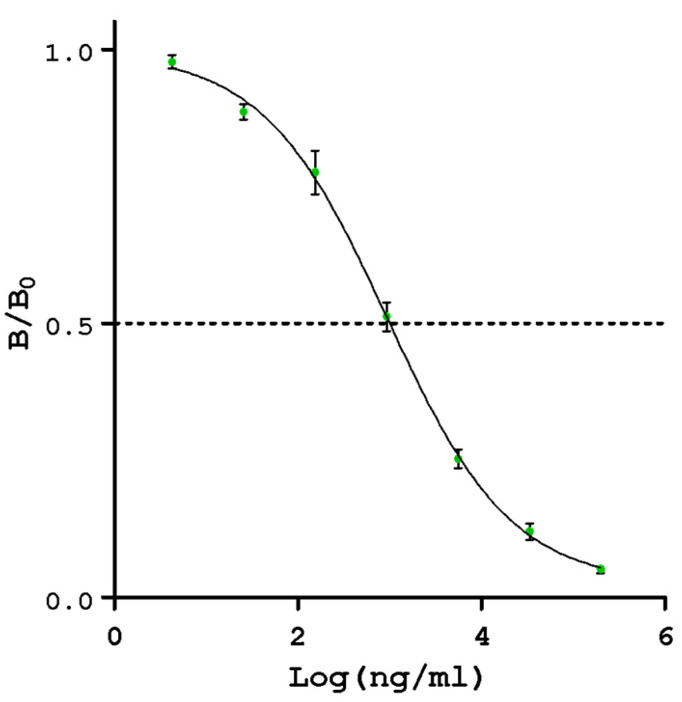
Calibration curve for BLG detection. Developed with whey solution in concentrations between 4 and 200,000 ng whey protein/mL, obtained by adjusting four logistic parameters (4PL). B/B0 is the absorbance of the sample divided by the absorbance in the absence of whey protein.

**Table 1 foods-12-03361-t001:** High hydrostatic pressure and enzymatic hydrolysis treatments applied to whey solutions.

Treatment ^†^		HHP ^•^	H ^⋄^	TIE ^⋆^	Note ^‡^
1:	C	-	-	-	Untreated WPC, control
2:	T-In	-	-	100 °C—10 min	WPC + thermal inactivation of the enzyme, TIE control
3:	HHP	300 MPa—5 min—45 °C	-	-	WPC + HHP
4:	H-In	-	45 °C—15 min + 25 °C—10 min	100 °C—10 min	Enzymatic hydrolysis
5:	H-In-HHP	300 MPa—5 min—45 °C	45 °C—15 min	100 °C—10 min	Enzymatic hydrolysis followed by TIE + HHP treatment
6:	H_ext_-In	-	45 °C—30 min + 25 °C—10 min	100 °C—10 min	Extended enzymatic hydrolysis
7:	H+HHP-In	300 MPa—5 min—45 °C	45 °C—15 min	100 °C—10 min	Enzymatic hydrolysis treatment combined with HHP
8:	HHPaH-In	300 MPa—5 min—45 °C	45 °C—5 min **^∎^**	100 °C—10 min	HHP-assisted enzymatic hydrolysis treatment

^†^ Coding of treatments applied to samples 1 to 8. ^•^ High hydrostatic pressure. ^⋄^ Enzymatic hydrolysis. ^⋆^ Thermal inactivation of the enzyme. ^‡^ Treatment description. ^∎^ Assisted hydrolysis implies hydrolysis during pressurization. For the preparation of the samples and the applied tests, the following procedure was followed.

**Table 2 foods-12-03361-t002:** Four-parameters logistic model (4PL) regression model for BLG ELISA.

4PL Regression Parameters	
A	0.995
B	0.025
C	942
D	0.644
R^2^	0.9959
Concentration Range (ng whey protein/mL)	4–200,000

Regression parameters: A: maximum absorbance, B: minimum absorbance, C: IC50 (inhibitory concentration 50%), D: slope.

**Table 3 foods-12-03361-t003:** Validation parameters for the BLG ELISA method •.

Parameter	Value
Limit of detection (ng whey protein/mL) (n = 24)	6.6 ± 0.02
Limit of quantification (ng whey protein/mL) (n = 24)	94.3 ± 0.05
Intra-assay variability (n = 24), CV (%)	8.0–11.9
Inter-assay variability (n = 27), CV (%)	7.53–12.5
Recovery (7 ng/mL, n = 5) (%)	70 ± 8.8
Recovery (95 ng/mL, n = 5) (%)	135 ± 10.6
Recovery (925 ng/mL, n = 5) (%)	98 ± 12.5

• Recovery values are expressed as mean ± coefficient of variation (CV).

**Table 4 foods-12-03361-t004:** Characterization of processed whey using combined hydrolysis treatments with bromelain and high hydrostatic pressure.

Treatment ^†^		%DH ^‡^	%BLG •
1:	C	0.00 ± 0.44 a	100.0 ± 4.8 B
2:	T-In	−0.21 ± 0.11 a	129.7 ± 7.2 A
3:	HHP	−0.55 ± 0.43 a	122.8 ± 0.7 A
4:	H-In	4.05 ± 0.31 b	86.6 ± 1.7 C
5:	H-In-HHP	4.18 ± 0.50 b	82.4 ± 8.6 C
6:	Hext-In	7.69 ± 0.45 c	72.9 ± 5.1 D
7:	H+HHP-In	8.20 ± 0.45 d	43.2 ± 1.3 E
8:	(HHPaH)-In	9.00 ± 0.40 d	20.7 ± 1.0 F

^†^ Coding of the treatments applied to samples 1 to 8, Table 1. ^‡^ Percentage of the degree of hydrolysis determined by the OPA method. Means with different lowercase letters are significantly different (*p* ≤ 0.05; Tukey). • Percentage of residual antigenicity of BLG (compared to untreated WPC) determined by ELISA. Means with different uppercase letters are significantly different (*p* ≤ 0.05; Tukey).

## Data Availability

The data used to support the findings of this study can be made available by the corresponding author upon request.

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
