# Peer review of "Assessment of the Efficiency of Technological Processes to Modify Whey Protein Antigenicity"

_foods, 2023, doi:10.3390/foods12183361_

Round 1

Reviewer 1 Report

This is a very well written article. The preparation of the whey fractions is meaningful. 

Only questions are raised, when reading the section with the rabbit immunisation (one or more rabbits?!) and the conclusions drawn from the experiments in the discussion. Even 20% residual antigenicity may include epitopes relevant for the allergy patients. This should be reflected and discussed with respect to next steps in the validation of the treatment methods. If serum only from one rabbit was used in the study, even more care should be taken with respect to the lack of variety of antibodies in the test. 

Author Response

This is a very well written article. The preparation of the whey fractions is meaningful.

R: Thank you for your positive comment. The detailed process for the whey fraction was included in the text (line 86).

Only questions are raised, when reading the section with the rabbit immunisation (one or more rabbits?!) and the conclusions drawn from the experiments in the discussion. Even 20% residual antigenicity may include epitopes relevant for the allergy patients. This should be reflected and discussed with respect to next steps in the validation of the treatment methods. If serum only from one rabbit was used in the study, even more care should be taken with respect to the lack of variety of antibodies in the test. 

R: Thank you for your suggestions, we included the following detail 

Line 143: Polyclonal antibodies were raised in trhee New Zealand White rabbits at the Central Animal Laboratory of FCEyN-UBA, Argentina. 

Line 156: Immune sera were separated by centrifugation, pooled -to conform a common polyclonal serum raised against denatured BLG-, diluted 1:1 in glycerol and then stored at -20 °C until further use

Also, we included the  following comment in line 572

In practical terms, the fact that the residual antigenicity can be evaluated directly or with minimal dilution minimizes the error associated with the high levels of dilution that would be necessary with commercial kits. It is important to consider that the aim of the study was to assess the ability of ELISA methods to detect modifications of BLG in whey protein when different treatments are applied. However, it is important to bear in mind that even 20% residual antigenicity may include epitopes relevant for allergy patients. Therefore, further studies would be necessary to biologically validate this aspect. In this regard, we are currently conducting more sophisticated studies in our lab and in collaboration with other groups, including the use of animal models.

Reviewer 2 Report

The authors investigated the efficiency of technological processes to modify whey protein antigenicity. The findings observed in this paper showed that enzymatic hydrolysis together with high hydrostatic pressure was the most effective combination to reduce whey protein solutions antigenicity. I think that the obtained results will be useful. Major revisions were attached.

1. Some results are included in the abstract to support the main findings.

2. Many sentences are missing references such as other technologies like enzymatic hydrolysis (EH) would be preferred because of their simplicity and low cost, constituting an economic alternative to modifying protein antigenicity (Lines 58-60).…………….and other sentences.

3. The introduction is not well organized, and lacks a sequential in the information to reach the objective and merit of study. The author should rewrite and cover aspects of the whey protein antigenicity.

4. Line 159: Mg2Cl > MgCl2

5. Line 177 and 180: Replace the references with original references for sensitivity such as “LOD is defined as the lowest concentration of antigen that can be distinguished from a sample not containing any antigen (B0), and is calculated with equation 3” and “The LOQ is calculated by subtracting ten times the standard deviation from the mean value of B0 and calculating the corresponding concentration from the equation of the respective calibration curve (equation 4)”.

6. Line 293: [25 > [25]

7. The full-name for abbreviations in the figure and table such as LOD, LOQ, DH, BLG ……………….should be added.

8. Add the ± SD or ± SEM in Table 3.

9. “” in Table 3 means …? This information should be added.

10. Recovery data should be provided.

11. References: Some extra references should be included in the text and discussed:
-Garcia-Mora P, Penas E, Frias J, et al. High-pressure improves enzymatic proteolysis and the release of peptides with angiotensin I converting enzyme inhibitory and antioxidant activities from lentil proteins. Food Chemistry, 2015, 171: 224-232.

-Espinosa-Pesqueira, Diana, Hernández-Herrero, et al. High Hydrostatic Pressure as a Tool to Reduce Formation of Biogenic Amines in Artisanal Spanish Cheeses. Foods, 2018, 7: 137.

-Segura-Gil I, A Blázquez-Soro, P Galán-Malo, et al. Development of sandwich and competitive ELISA formats to determine β-conglycinin: Evaluation of their performance to detect soy in processed food. Food Control, 2019, 103:78-85.

Author Response

  1. Some results are included in the abstract to support the main findings.

R: Thank you for your suggestion. We interpret that you mean that additional results should be included in the abstract to support the main findings.

  1. Many sentences are missing references such as other technologies like enzymatic hydrolysis (EH) would be preferred because of their simplicity and low cost, constituting an economic alternative to modifying protein antigenicity (Lines 58-60).…………….and other sentences.

R: Thank you for your suggestion, and for providing us with some additional references (item 11). We have included some additional references (lines 38, 47, and 51)

Other technologies such as enzymatic hydrolysis (EH) would be preferred because of their simplicity and low cost, representing an economic alternative to modifying protein antigenicity [7,8]. In fact, whey hydrolysates have been used for a long time in infant nutrition as substitutes for human breast milk [9]. The selection of hydrolysis methods mainly depends on the protein source. For example, keratin sources such as feathers, horns, and beaks usually require acidic or alkaline hydrolysis treatments, or the use of bacterial keratinases. On the other hand, the proteolysis of animal products such as whey and meat, or plant ingredients such as soy and legume proteins, often requires enzymatic or microbial hydrolysis [10]. In this regard, different peptidases also allow the production of oligopeptides with different biological properties. The identification of new enzymes, especially of microbial origin, able to act with different specificities over different protein sources represents an active field of research [11].

  1. Wróblewska, B.; Karamać, M.; Amarowicz, R.; Szymkiewicz, A.; TroszyÅ„ska, A. & Kubicka, E. Immunoreactive properties of peptide fractions of cow whey milk proteins after enzymatic hydrolysis. Int. J. Food Sci. Technol. 2004, 39(8), 839–850.
  2. Bhat, Z. F.; Kumar, S. Bioactive peptides from egg: a review. Nutr. Food Sci. 2015, 45, 190–212. doi: 10.1108/NFS-10-2014-0088
  3. Salvatore, S.; Acunzo, M.; Peroni, D.; Pendezza, E.; Di Profio, E.; Fiore, G.; Zuccotti, G. V, & Verduci, E. Hydrolysed Formu-las in the MRanagement of Cow’s Milk Allergy: New Insights, Pitfalls and Tips. Nutrients. 2022, 13(8), 2762. doi: 10.3390/nu13082762
  4. da Silva, R.R. Enzymatic Synthesis of Protein Hydrolysates From Animal Proteins: Exploring Microbial Peptidases. Front. Microbiol. 2018, 9, 1-5. doi:0.3389/fmicb.2018.00735
  5. da Silva, R. R. Comment on mRNA-Sequencing analysis reveals transcriptional changes in root of maize seedlings treated with two increasing concentrations of a new biostimulant. J. Agric. Food Chem. 2018, 66, 2061–2062. doi: 10.1021/acs.jafc.8b00022
  6. The introduction is not well organized, and lacks a sequential in the information to reach the objective and merit of study. The author should rewrite and cover aspects of the whey protein antigenicity.

R: Thank you for your comment. We have reorganized the introduction to give sequential information, and also added some references.

  1. Line 159: Mg2Cl > MgCl2

R: Thank you for finding this mistake. It was corrected.

  1. Line 177 and 180: Replace the references with original references for sensitivity such as “LOD is defined as the lowest concentration of antigen that can be distinguished from a sample not containing any antigen (B0), and is calculated with equation 3” and “The LOQ is calculated by subtracting ten times the standard deviation from the mean value of B0 and calculating the corresponding concentration from the equation of the respective calibration curve (equation 4)”.

R: Thank you for your comment. We have also added precision and accuracy information

  1. Line 293: [25 > [25]

R: Thank you for finding this mistake. It was corrected.

  1. The full-name for abbreviations in the figure and table such as LOD, LOQ, DH, BLG ……………….should be added.

R: Thank you for your suggestion. The meaning of abbreviations has been added.

  1. Add the ± SD or ± SEM in Table 3.

R: The required additional information has been added.

  1. “‡” in Table 3 means …? This information should be added.

R: Thank you for the suggestion. Modifications were made in Table 3, and the information has been added.

  1. Recovery data should be provided.

R: Thank you for the suggestion. We have included some validation details as follows (line 363):

Accuracy

Five samples with known concentrations of the calibrator were spiked in duplicate at the recovery values considered most relevant: an uncontaminated sample (0 ng/ml), a sample at a value close to the LOD (7 ng/ml), a sample at a value close to the LOQ (95 ng/ml), a sample at the value considered as acceptance, and a sample at a value close to 50% of the curve (925 ng/ml). The recovery in all cases was between 70 and 135%, expressed as a percentage of the analyte. These results were consistent with similar studies [28]

Although different food commodities should be included in cross-reactivity testing for ELISA method validation, as stated by AOAC [29], this validation was not carried out in the present study because the use of the ELISA was not intended for the detection of milk traces in different food matrices

  1. References: Some extra references should be included in the text and discussed:

R: We deeply appreciate your provision of these relevant references. We have included them along the text

Line 332 and 338- [28]: Segura-Gil I, A Blázquez-Soro, P Galán-Malo, et al. Development of sandwich and competitive ELISA formats to determine β-conglycinin: Evaluation of their performance to detect soy in processed food. Food Control, 2019, 103:78-85.

Line 545- [48]: Espinosa-Pesqueira, Diana, Hernández-Herrero, et al. High Hydrostatic Pressure as a Tool to Reduce Formation of Biogenic Amines in Artisanal Spanish Cheeses. Foods, 2018, 7: 137.

Line 553- [49]: Garcia-Mora P, Penas E, Frias J, et al. High-pressure improves enzymatic proteolysis and the release of peptides with angiotensin I converting enzyme inhibitory and antioxidant activities from lentil proteins. Food Chemistry, 2015, 171: 224-232.

Reviewer 3 Report

In this manuscript, the authors used laboratory-developed cELISA to evaluate the BLG immunoreactivity under different treatments. I have the following major concerns. First, the authors proposed the immunoreactivity changes of BLG and gave some explanations. To me, the authors did not use secondary methods such as commercial ELISA kits or immunoblot to verify their findings. Using lab-developed cELISA without validation could give false detection results. Second, during the ELISA, there were many variables that could lead to the different immunoreactivity. For example, the authors did not give any sample loading information, is it loading by volume or by mass? How to control the protein mass difference in ELISA? Third, the written English of this manuscript must be improved. There were many inconsistency throughout the manuscript. Many sentences are very confusing to read. Please see my other comments below:

L43-45: Which sample? If the sample contained high quantity of milk, why needs to be detected?

L49: Give the full name of MW, BLG, AA, etc. when they first showed up in the manuscript.

L51: Which concentration is >50 M?

L61: episodes or epitopes?

L86-87: Grammar mistake.

L98-104: Those explanations could be moved to the results and discussion section since they were not related to methods.

L121: Keep the unit consistent.

L123: Keep the buffer formula consistent, sometimes the authors used / between two ingredients. Also, the authors gave the formula of TBS in L125 but not the formula of borate buffer.

L130: Why not use the Sigma BLG solution as immunogen?

L145-146: The calibration curve of ELISA? Why specifically mention about 0.5% WPC? The concentration between 0.026 and 2000 already indicated the protein concentration of WPC.

L148-149: Why did the authors prepare a total of 200 µL reagent in the competition. The coating volume was only 100 µL, the remaining height has been blocked.

L168: Is it EC50 or IC50? In a competitive ELISA, shouldn’t be concentration at half-maximal inhibition?

Fig. 1 is problematic. The x-axis needs to be changed to the log scale. Where did the 4 come from? The authors mentioned 0.026 in the methodology. No sample size information as well.

L264: Have no clue the purpose of Table 2. No explanation and confusing caption.

L269-270: Inter or intra?

Table 3: It seems to be unnecessary. All those values have been reported in the context. Also, this table is very confusing. On the left column, it seems the authors want to report the LOD and LOQ as ng whey proteins/mL, but in the right column, the authors reported as BLG. It also has a superscript without explanation.

L282-283: The usage of polyclonal antibodies in ELISA is not equivalent to IgE immunoreactivity. Therefore, I question the comparison of ELISA validation results with the immunologically active proteins. In addition, how to convert the ELISA unit (µg/mL) to µg antigen/g protein?

L287: How to convert to 10 mg protein/mL?

L293: Not correctly cited.

The written English of this manuscript must be improved. There were many inconsistency throughout the manuscript. Many sentences are very confusing to read.

Author Response

-In this manuscript, the authors used laboratory-developed cELISA to evaluate the BLG immunoreactivity under different treatments. I have the following major concerns.

-First, the authors proposed the immunoreactivity changes of BLG and gave some explanations. To me, the authors did not use secondary methods such as commercial ELISA kits or immunoblot to verify their findings. Using lab-developed cELISA without validation could give false detection results.

R: We understand your concern. We have been working on this topic for several years. Along this time, we have developed several methods, most of them with antibodies produced in our lab, raised against different immunogens. We have tried specific protein markers of milk or whey, such as BLG, ALA, or the entire whey. We have also worked with commercial kits, mainly from r-Biopharm and Neogen. In the development of these methods, we conducted several correlation studies, comparing for instance different in-house produced antibodies with each other, or with the commercial kits, and in all cases we obtained correlation coefficients higher than 0.9. This has given us the experimental support to use the present method, which was the one that performed best for the intended use. In addition, it would be difficult to compare in this study the developed method with commercial kits, since they were developed for a different intended use (i.e. to detect traces amounts of milk or whey as a contaminant). For instance, to attain low LOD and LOQ, it is usual to employ the peroxidase as a conjugated enzyme, while in the present study, we used alkaline phosphatase, since our objective was to attain higher levels of working range, but with lower dilutions. According to our previous experiments, we found a high variability when using commercial kits to assess the effect of different treatments on WPC antigenicity.

-Second, during the ELISA, there were many variables that could lead to the different immunoreactivity. For example, the authors did not give any sample loading information, is it loading by volume or by mass? How to control the protein mass difference in ELISA?

R: We understand that the production of HIF is a very particular topic, since the ultimate objective is to produce an ingredient (protein hydrolysate) able to preserve the nutritional quality of the protein (it could be equivalent to the protein mass), but with a diminished immunologic reactivity (it could be equivalent to clinical performance of the protein ingredient). The translation of one of this aspect into the other represents a true challenge, and here is where some aspects can be confusing, since the clinical standard of a HIF is, literally, that clinically hypoallergenic formulas must not cause allergic symptoms when fed to 90% of infants allergic to the base protein used to manufacture the formula, which is impossible to measure at the manufacturing site. However, it would be possible to do an immunochemical assessments of the antigen content (what was accomplished in the present study) and to predict the allergenic potential of modified protein systems using much more standardized animal models of immunogenicity (what is currently underway in our lab, in collaboration with another scientific group and is planned to be published in future papers). In words of Cordle (2006), ”The questions to be made are: How much reduction in immunologic reactivity is sufficient to achieve hypoallergenic clinical performance?, which is analogous to the key allergen question in the general food industry, How much unlabeled food allergen cross-contamination is too much?”. Data from HIF clinical studies indicate that the equivalent to 0.2 mg of milk allergen (actually immunologically active antigen) per feeding might be well in the hypoallergenic range. Bearing this explanation in mind, for the calibration curve, we used a whey solution, and with this condition, the ELISA reading can be translated into true amount of protein. However, once the hydrolytic process begins, what is measured is the equivalent of “immunologically active antigen”, which can not be translated anymore to amount of protein. So, during the monitoring of the hydrolytic process, we are not actually controlling the difference in protein mass, because it is supposed to be the same, but the difference (hopefully the decrease, although for some treatments, such as heat treatments, it was the opposite in antigenicity terms) of the antigenicity equivalent of the sample. It is true that we are not measuring allergenicity, which has a clinical connotation, but the antigenicity, which is the best approximation we can do at a manufacturing site, and that can represent a control parameter for the production process.

  1. Cordle, Chapter 16 - Allergen quality assurance for hypoallergenic formula, Editor(s): Stef J. Koppelman, Sue L. Hefle, Detecting Allergens in Food, Woodhead Publishing, 2006, Pages 293-314, ISBN 9781855737280, https://doi.org/10.1533/9781845690557.4.293.

-Third, the written English of this manuscript must be improved. There were many inconsistency throughout the manuscript. Many sentences are very confusing to read.

R: Thank you for your remark. The grammar mistakes were corrected, several sentences were rewritten to make them more readable, and the corrected version was proofread by a native speaker.

-Please see my other comments below:

L43-45: Which sample? If the sample contained high quantity of milk, why needs to be detected?

R: We understand your concern. Actually, this is related to one of the objectives of the present study. For the production of whey hydrolysates to be used in HIF, the final product must meet a specification standard, which some researchers report to be between 1 and 15 ug/ml. This could be achieved with a commercial kit. However, at the development stage, it would be necessary to monitor the process, from the beginning to the end, to select the best technological alternative, or to make any improvement. This is where the commercial kit could fail, since they are developed to detect contaminant traces in foods, and not to quantify relatively higher concentrations, such as those present in a partially hydrolyzed sample. Therefore, our objective is to provide a potential user with an analytical tool to monitor the process at any stage.

This was transcribed from Cordle, 2006: “As a general rule for formulas based on cows’ milk protein hydrolysates, hypoallergenic clinical performance can be achieved at residual allergen concentrations below 15 μg/mL, with the most effective formulas containing less than 1 ug allergen/mL. (Testing is performed at ready-to-feed concentrations on liquid formula or reconstituted powdered formula, serving size is approximately 250 mL.).”

Cordle, Chapter 16 - Allergen quality assurance for hypoallergenic formula, Editor(s): Stef J. Koppelman, Sue L. Hefle, Detecting Allergens in Food, Woodhead Publishing, 2006, Pages 293-314, ISBN 9781855737280, https://doi.org/10.1533/9781845690557.4.293.

(https://www.sciencedirect.com/science/article/pii/B9781855737280500165)

L49: Give the full name of MW, BLG, AA, etc. when they first showed up in the manuscript.

R: Thank you for your comment. The meaning of the different abbreviations were included.

L51: Which concentration is >50 M?

R: Thank you for your remark. It was changed to 50 µM of BLG

L61: episodes or epitopes?

R: We confirm that it was referred to clinical allergy episodes (allergy reactions underwent by allergic people).

L86-87: Grammar mistake.

R: Corrected in the revised version.

L98-104: Those explanations could be moved to the results and discussion section since they were not related to methods.

R: Thank you for your comment. It was rewritten, and the information that could be considered as Results was accordingly removed

L121: Keep the unit consistent.

R: Thank you for your observation.  The text was carefully revised according to this remark.

L123: Keep the buffer formula consistent, sometimes the authors used / between two ingredients. Also, the authors gave the formula of TBS in L125 but not the formula of borate buffer.

R: Thank you for your remark. The required information was accordingly changed or provided in the revised version.

L130: Why not use the Sigma BLG solution as immunogen?

R: We understand your concern. Actually, we did use sigma BLG to standardize and make our results easier to replicate for other researchers. However, we did not use it directly, but first we applied a denaturation treatment, to use it as an immunogen. This was done taking into account that the treatments to be applied would denature BLG, and therefore, we would increase the detectability of the whey protein and peptide thereof. This is a strategy also used in commercial kits, typically those developed to detect egg traces in food, where antibodies were raised against heat treated egg proteins. 

L145-146: The calibration curve of ELISA? Why specifically mention about 0.5% WPC? The concentration between 0.026 and 2000 already indicated the protein concentration of WPC.

R: The protein concentration was corrected. It raged from 4 to 200,000 ng/ml

L148-149: Why did the authors prepare a total of 200 µL reagent in the competition. The coating volume was only 100 µL, the remaining height has been blocked.

R: Thank you for the observation. There was a typing mistake in the protocol; 200 ul/well was used for coating plates, and 250 ul/well for blocking. It was corrected with the right information.

L168: Is it EC50 or IC50? In a competitive ELISA, shouldn’t be concentration at half-maximal inhibition?

R: Thank you for your observation. Indeed, for competitive ELISA is IC. We have also modified the expression of the formula to make it easier to understand.

Fig. 1 is problematic. The x-axis needs to be changed to the log scale. Where did the 4 come from? The authors mentioned 0.026 in the methodology. No sample size information as well.

R: Thank you for your comment. Fig 1: It is stated in the figure that the x axes is log (Conc). The concentrations used to prepare calibration curves were from 4 to 200,000 ng protein/ml.

L264: Have no clue the purpose of Table 2. No explanation and confusing caption.

R Caption was rewrote

Using the data obtained from the different concentrations of the calibrator, the standard curve was constructed according to the Four parameters logistic model (4PL) mathematical regression model for BLG ELISA, which has the following parameters:

  • A (maximum absorbance): 0.0995
  • B (minimum absorbance): 0.025
  • C (IC50 – Inhibitory Concentration 50%): 942
  • D (slope): 0.644

The concentration range of the curve is 4 – 200000 ng whey protein/ml. The curve has a high R2 of 0.9959.

L269-270: Inter or intra?

R: Validation information was added. Precision information for intra and interassay was described

Table 3: It seems to be unnecessary. All those values have been reported in the context. Also, this table is very confusing. On the left column, it seems the authors want to report the LOD and LOQ as ng whey proteins/mL, but in the right column, the authors reported as BLG. It also has a superscript without explanation.

R: Table 3was modified, and validation parameters were included

L282-283: The usage of polyclonal antibodies in ELISA is not equivalent to IgE immunoreactivity. Therefore, I question the comparison of ELISA validation results with the immunologically active proteins. In addition, how to convert the ELISA unit (µg/mL) to µg antigen/g protein?

R: As previously explained, one of the main concerns in HIF production is how much reduction in immunologic reactivity is sufficient to achieve hypoallergenic clinical performance. This is analogous to the question in the general food industry, "How much unlabeled food allergen cross-contamination is too much?". Commercial kits (which are designed to detect and quantify allergen traces) measure protein traces, which can be equivalent to mass of allergenic protein. In the case of protein hydrolysates, ELISA measures immunologically active antigen, which can be regarded as a production control parameter or specification control parameter of the final product (hydrolysate or HIF). This can be used to estimate the remaining immunologically active antigen in the HIF, which is what is clinically relevant for the allergic person.

L287: How to convert to 10 mg protein/mL?

R: As previously explained, even though that it is true that we are not measuring allergenicity (IgE immunoreactivity is not equivalent to ELISA results, were IgG is measured), which has a clinical connotation, the antigenicity is the best approximation we can do at a manufacturing site, and that can represent a control parameter for the production process. In this regard, 10 mg protein/ml are typical values reported in literature.

L293: Not correctly cited.

R: Thank you for the comments. It was accordingly corrected.

Comments on the Quality of English Language

The written English of this manuscript must be improved. There were many inconsistency throughout the manuscript. Many sentences are very confusing to read.

R: Thank you for your remark. Regarding the English language, we have corrected the grammar mistakes and proofread it by a native speaker.

Round 2

Reviewer 2 Report

All issues have been solved.